# Design, Synthesis, and Photophysical Properties of 5-Aminobiphenyl Substituted [1,2,4]Triazolo[4,3-*c*]- and [1,2,4]Triazolo[1,5-*c*]quinazolines

**DOI:** 10.3390/molecules29112497

**Published:** 2024-05-24

**Authors:** Tatyana N. Moshkina, Alexandra E. Kopotilova, Marya A. Ivan’kina, Ekaterina S. Starnovskaya, Denis A. Gazizov, Emiliya V. Nosova, Dmitry S. Kopchuk, Oleg S. El’tsov, Pavel A. Slepukhin, Valery N. Charushin

**Affiliations:** 1Organic and Biomolecular Chemistry Department, Chemical Technology Institute, Ural Federal University, Mira St. 19, 620002 Ekaterinburg, Russia; tan.moshckina@yandex.ru (T.N.M.); kopotilova.alexandra@yandex.ru (A.E.K.); e.s.starnovskaia@urfu.ru (E.S.S.); dkopchuk@mail.ru (D.S.K.); o.s.eltsov@urfu.ru (O.S.E.); slepukhin@ios.uran.ru (P.A.S.); valery-charushin-562@yandex.ru (V.N.C.); 2Postovsky Institute of Organic Synthesis, Ural Branch of the Russian Academy of Sciences, S. Kovalevskaya Str., 22, 620108 Ekaterinburg, Russia; dengaz94@mail.ru

**Keywords:** [1,2,4]triazolo[4,3-*c*]quinazoline, [1,2,4]triazolo[1,5-*c*]quinazoline, cross-coupling, fluorescence, solvatochromism

## Abstract

Two series of novel [1,2,4]triazolo[4,3-*c*]- and [1,2,4]triazolo[1,5-*c*]quinazoline fluorophores with 4′-amino[1,1′]-biphenyl residue at position 5 have been prepared via Pd-catalyzed cross-coupling Suzuki–Miyaura reactions. The treatment of 2-(4-bromophenyl)-4-hydrazinoquinazoline with orthoesters in solvent-free conditions or in absolute ethanol leads to the formation of [4,3-*c*]-annulated triazoloquinazolines, whereas [1,5-*c*] isomers are formed in acidic media as a result of Dimroth rearrangement. A 1D-NMR and 2D-NMR spectroscopy, as well as a single-crystal X-ray diffraction analysis, unambiguously confirmed the annelation type and determined the molecular structure of *p*-bromophenyl intermediates and target products. Photophysical properties of the target compounds were investigated in two solvents and in the solid state and compared with those of related 3-aryl-substituted [1,2,4]triazolo[4,3-*c*]quinazolines. The exclusion of the aryl fragment from the triazole ring has been revealed to improve fluorescence quantum yield in solution. Most of the synthesized structures show moderate to high quantum yields in solution. Additionally, the effect of solvent polarity on the absorption and emission spectra of fluorophores has been studied, and considerable fluorosolvatochromism has been stated. Moreover, electrochemical investigation and DFT calculations have been performed; their results are consistent with the experimental observation.

## 1. Introduction

[1,2,4]Triazoloquinazoline represents polyazaheterocycle, which consists of triazolo moiety fused to a quinazoline ring. Its derivatives are widely known as an important class of heterocyclic aromatic compounds for various pharmaceutical applications [1,2]. Moreover, triazoloquinazolines provide a promising molecular platform for materials sciences. Each of the fragments is of great interest owing to their electron withdrawing properties and use in the design of donor–acceptor small molecules displaying characteristics preferable for optical materials. The quinazoline component has been explored in the context of fundamental research [3,4,5], with some quinazoline derivatives revealed to have potential application in optoelectronics [6,7,8,9], detection of analytes [10,11], bioimaging [12], etc. [1,2,4]Triazole derivatives, in turn, are considered as blue phosphorescent, TADF emitters or host materials for OLED devices [13,14,15]. Other D-π-A-π-D structures with a triazole ring as an acceptor part show strong emission in solution and potential for optoelectronic purposes [16,17]. Due to high photoluminescence (PL) efficiency, as well as good affinity to analytes, triazole-based materials have great potential to be used as sensitive and selective fluorescence probes [18].

The advantages of fused triazoles, such as the extended π-conjugation system, rigid planar stricture, tuning of the molecular energy levels, and good thermal and morphological stabilities were taken into account when creating the [1,2,4]triazolo[4,3-*a*]pyridine and [1,2,4]triazolo[1,5-*a*]pyridine luminophores [19,20,21]. Moreover, [1,2,4]triazolopyridine derivatives are suitable scaffolds for the designing of highly twisted π-conjugation structures, maintaining high triplet energies. This strategy was realized in the synthesis of 9,9′-(2-([1,2,4]triazolo[1,5-*a*]pyridin-2-yl)-1,3-phenylene)bis(9*H*-carbazole) host materials [22]; devices based on this heterocycle demonstrate remarkable electroluminescence performance comparable to that for reported Ir-based PhOLEDs.

Notably, the data concerning photophysical properties of [1,2,4]triazoloquinazoline derivatives are scarce. Some related compounds have been shown to be effective luminescent components for light-emitting devices [23,24,25,26,27,28]. Recently the [1,2,4]triazolo[5,1-*b*]quinazoline probe toward Fe^3+^ ions has been reported [29]. However, to date, there are no systematic structure–property relationship studies.

In this context, design, synthesis, and investigation of photophysical properties of triazolo-annulated quinazolines are highly important and interesting for field of both fundamental and applied chemistry. Previously, we have reported synthesis and photophysical properties of 3-aryl-substituted 5-(4′-amino[1,1′]-biphenyl)[1,2,4]triazolo[4,3-*c*]quinazolines **A**, Figure 1 [30]. Some of the obtained compounds were shown to exhibit strong fluorescence, both in solution and in solid state, as well as emission solvatochromism and sensory ability toward water and acid. It was revealed that the annulation of [1,2,4]triazole ring to the quinazoline core had a considerable impact on emission behavior and solvatochromic properties compared to 4-morpholinylquinazolines, 4-cyanoquinazolines, or quinazolin-4-one counterparts **B**, Figure 1 [31,32].

Herein, we aim to modify the triazole fragment and design of 5-(4′-amino[1,1′]-biphenyl)[1,2,4]triazolo[4,3-*c*]quinazolines **I** unsubstituted at position 3 and their 3-ethyl analogues. We suppose that excluding the aryl fragment from the triazole ring might have considerable influence on photophysical properties. Moreover, we are interested in whether the isomeric arrangement of the triazoloquinazoline ring will have a significant impact on photophysical characteristics; for this purpose, we developed [1,2,4]triazolo[1,5-*c*]quinazoline derivatives **II**. We used 2-(4-Bromophenyl)- 4-hydrazinoquinazoline and orthoester as starting materials for the construction of polycyclic [1,2,4]triazolo[4,3-*c*]quinazoline core of compounds **I**. Their [1,5-*c*] isomers were obtained in acidic media as a result of Dimroth rearrangement of [1,2,4]triazolo[4,3-*c*]quinazolines. The cross-coupling of bromophenyl derivatives with boronic acids under the typical conditions was applied for the synthesis of target fluorophores. Photophysical and electrochemical properties for compounds **I** and **II** were carefully studied experimentally and theoretically using DFT calculations. Additionally, characteristics of target fluorophores **I** and **II** and their 3-aryltriazolo[4,3-*c*]quinazoline counterparts **A** were compared.

## 2. Results

### 2.1. Synthesis

The synthetic approach (Figure 1) is based on the use of previously described 2-(4-bromophenyl)-4-hydrazino-quinazoline **1** [33] as starting material. [1,2,4]Triazolo[4,3-*c*]quinazolines **2a**,**b** were prepared by solvent-free cyclocondensation of **1** with triethyl orthoformate or triethyl orthopropionate under reflux for 4 h in good yields of 83 and 84%, respectively. A similar procedure was described previously for related compounds [34]. It was shown that the refluxing of staring hydrazine **1** with orthoesters in anhydrous ethanol gives triazolo[4,3-*c*]quinazolines **2a**,**b**, with comparable yields. However, using 95% ethanol as a solvent resulted in the mixture of isomers **2** and **3** and the ring-opening product **3b′**. The compounds **2a**,**b** were successfully converted into triazolo[1,5-*c*]quinazoline isomers **3a**,**b** by refluxing in glacial acetic acid for 4 h. The reaction progress can be easily monitored by TLC analysis. The R_f_ values of isomers are significantly different (for example, 0.16 for **2a** and 0.71 for **3a** in a 1/1 mixture of hexane/EtOAc).

The transformation, probably, is based on an H^+^-catalyzed Dimroth rearrangement (Appendix A) proposed for a similar [1,2,4]triazolopyrimidine heterocycle; the mechanism of this process generally involves the addition of an electrophile, a ring opening, and a ring closure [35]. Each isomer **2a** and **3a** or **2b** and **3b** was distinguished by their ^1^H NMR spectra (Appendix A). For example, the most prominent peak in the spectrum of compound **2a** was observed at 9.32 ppm as a singlet attributed to the triazole proton, while a similar singlet in the spectrum of isomer **3a** was observed upfield at 8.63 ppm, Figure 2a. In the case of the ethyl-substituted derivatives **2b** and **3b**, there is considerable difference in the position of signals attributed to ethyl group, Figure 2b. Moreover, the signals of phenylene protons (H-2 and H-6) of [1,5-*c*] isomers **3a**,**b** shifted downfield compared to [4,3-*c*] ones **2a**,**b**, which indicates an increase in the electron-withdrawing effect of the annulated triazole cycle. The NMR correlations are in good agreement with the literature data for triazolo-annelated azacycles [36,37]. Additionally, we performed a nuclear Overhauser effect spectroscopy (NOESY) and heteronuclear multiple bond correlation experiment (HMBC) for compounds **2a** and **3b** (Appendix A). In the ^1^H-^1^H NOESY spectra of compound **2a** we observed a cross-peak between H-3 and H-2′ proton signals, whereas correlations with triazole proton in spectrum of compound **3a** did not appear. ^1^H-^13^C HMBC spectrum of **2a** contains a cross-peak of the C(5) atom with an H-3 proton of the triazole cycle; in the case of its isomer **3a**, the corresponding cross-peak is absent. These findings are consistent with proposed structures and confirm spatial arrangement of molecules. Notably, the melting point of [4,3-*c*] isomers **2a**,**b** is higher than that of [1,5-*c*] counterparts **3a**,**b** (for example, 250–252 °C for **2a** and 186–188° for **3a**, respectively).

Furthermore, it was established that both compounds **3a** and **3b** can be obtained directly from 2-(4-bromophenyl)-4-hydrazinoquinazoline **1** by treatment with orthoester in acidic media; the refluxing of the reaction mixture for 16 h generated the desired products in 91% and 89% yields, respectively, after recrystallization from DMSO. The reaction progress was monitored by NMR spectroscopy after 4, 8, and 16 h, Appendix A. Each time after cooling the reaction mixture to room temperature, water was added and the precipitate that formed was filtered off, dried, and analyzed. The spectroscopic data shows that no signals of the starting compound that has been observed at 4 h in both cases, whereas signals of both isomers, as well as ring-opening product **3b′**, appear in the first case (Appendix A). Probably, the formation of triazolo[1,5-*c*]quinazoline proceeds through [4,3-*c*] isomers with subsequent rearrangement. After 16 h the ring-opening products were fully converted into corresponding triazolo[1,5-*c*]quinazolines. It is worth noting, we succeeded in isolating the ring-opening product **3b′** (Figure 1), which seems to be the intermediate during 5-(4-bromophenyl)-2-ethyl-[1,2,4]triazolo[1,5-*c*]quinazoline formation, by column chromatography. The amide **3b′** probably formed as a result of the hydrolysis; the structure of proposed compound **3b′** is consistent with the ^1^H NMR spectroscopy and the mass spectrometry data (Appendix A).

Both isomers participate in cross-coupling reactions under typical conditions described elsewhere [31,38,39] and form products **4a**–**f** or **5a**–**f** in moderate to good yields (from 36 to 77% after purification by column chromatography on silica gel or recrystallisation from DMSO). Their structure was confirmed by spectroscopic and analytical data. Notably, the NMR spectra of products **4** and **5** are significantly different depending on annelation type, similar to their parent bromo derivatives **2** and **3**, (Appendix A). To obtain an unambiguous structural assignment of each isomer, we grew single crystals of **4a** and **4e**, as well as **5d** and **5e**, for X-ray diffraction analysis (Figure 3, Appendix A). Single crystals were obtained by the slow evaporation technique from an *n*-hexane/CH_2_Cl_2_ mixture for **4a**, **4e**, and **5d** or an n-hexane/EtOAc mixture for **5e**.

According to XRD data, the compounds are crystallized in the centrosymmetric space groups of the monoclinic or triclinic systems. The general geometry, bond distances, and angles of the compound are near to expectations. In particular, the nitrogen atom of the diethylaminophenyl or diphenylaminophenyl substituents has a planar configuration with neighboring carbon atoms. The triphenylamino group of compounds **4e** and **5e** is twisted and has a propeller-like shape. The compounds **4a** and **5e** demonstrate the disordering of the ethyl groups. All the compounds are characterized by twisted conformation of the biphenylene moiety around a heteroaromatic core with the highest torsion angle in **4e** N(5)C(13)C(15)C(20) = 83.8°. For other studied compounds, the torsion angle between the heterocycle and phenylene substituents is significantly lesser due to the effect of the π-π conjugation. In the crystals the shortened intermolecular π-π contacts are observed. For compound **5d**, the contact C(9)…C(20) [x − 1, y, z] 3.253 Å between π-accepted heterocyclic and π-donated biphenylene moieties was noticed. The compound **5e** forms the π-interacted centrosymmetric dimers with distance C(14)…C(16) [1 − x, 1 − y, 1 − z] 3.329 Å. For compound **4e**, π-π interaction is observed between heterocyclic moieties [2 − x, −y, 1 − z] with a distance of 3.24 Å between the least-squared planes. For compound **4a** most principal intermolecular contacts are weak H-bonds C(4)H…N(5) [1 − x, −y, 1 − z] contributing to the formation of H-bonded dimers. However, the π-π interactions for this compound are insignificant.

### 2.2. UV/Vis and Fluorescence Spectroscopy

The UV/Vis absorption and photoluminescence (PL) spectroscopic data of [1,2,4]triazolo[4,3-*c*]quinazolines **4a**–**f** and [1,2,4]triazolo[1,5-*c*]quinazolines **5a**–**f** were studied for toluene and MeCN solutions at c ~10^−5^ M and presented in Table 1; the corresponding spectra are shown in Appendix A.

Normalized absorption spectra of compounds **4a**–**f** and **5a**–**f** in MeCN are combined in Figure 4. As can be seen, the lowest energy absorption maxima are affected by the nature of the aminoaryl fragment. 9*H*-Carbazol-9-yl-containing triazoloquinazolines **4c**, **4f**, **5c**, and **5f** are characterized by similar absorption features and display maxima in the range of 312–328 nm, whereas the absorption band of their Et_2_N or Ph_2_N counterparts is red-shifted, and the maxima are located in the range of 340–375 nm. On the other hand, the presence of the Et group at the triazole ring of [1,2,4]triazolo[4,3-*c*]quinazolines results in hypsochromically-shifted absorption (compounds **4d**, **4e**, and **4f** in contrast to **4a**, **4b** and **4c**) that can be associated with considerable twisting of biphenyl moiety influenced by the ethyl group. In the case of [1,2,4]triazolo[1,5-*c*]quinazolines **5a**–**f** the presence of ethyl substituent at triazole skeleton has little effect on the absorption wavelength (pairs **5d**–**5a**, **5e**–**5b**, and **5f**–**5c** in Table 1 and Figure 4b). The [1,5-*c*] annelation type, in general, leads to a shift of the absorption maxima in the red region compared to the [4,3-*c*] one.

Compared to the MeCN solution, the absorption band in toluene is slightly shifted to the red region, Table 1, but, in general, the influence of solvent polarity on the absorption band is minor.

All the compounds of **4** and **5** are emissive in both solvents with different fluorescence intensity and quantum yield. The emission maxima of Et_2_N- and Ph_2_N-substituted triazoloquinazolines are in the range of 465–486 nm in toluene, whereas carbazolyl-containing triazoloquinazolines emit in the blue–purple region with maxima at 420–441 nm. The influence of the arrangement of the triazole core, as well as the presence of the Et-group, is negligible, Appendix A. However, all fluorophores are found to be sensitive in response to the polarity of the solvent. When going from toluene to MeCN, the emission band shifts to the red region and the maxima appear at 530–548 nm in the case of carbazolyl-derivatives **4c**, **4f**, **5c**, and **5f**, and in the range of 593–609 nm for other counterparts, **4a**,**b**,**d**,**e** and **5a**,**b**,**d**,**e**. According to the obtained quantum yields in the two solvents, compounds can be divided into several groups, namely: compounds **5d** and **5f** with Φ_F_ ≥ 90% in both solvents; compounds **4a**, **4b**, **5a**, **5b**, and **5e** with Φ_F_ ≥ 90% in toluene (non-polar media); compounds **4c**, **4d**, **4e**, and **5c** with moderate Φ_F_ (11–75%, depending on solvent); and compound **4f** with low emission, less 3%. Moreover, the compounds **4a**, **4b**, and **5a**–**f** demonstrate a decrease in quantum yield when going from non-polar toluene to polar MeCN whereas the compounds **4c**, **4d**, **4e**, and **4f** show enhancement emission in polar media compared to a non-polar one.

For detailed investigation of photophysical properties we measured the fluorescence lifetime (τ) of chromophores **4a**–**d** and **5a**–**f** in toluene (Appendix A), and also calculated radiative decay rate constant (k_r_) and non-radiative decay rate constant (k_nr_) (Table 2). Emission spectra for fluorophores **4a**,**b**,**d** and **5a**–**f** fit the single exponential function, whereas decay trace is bi-exponential in the case of compounds **4c** and **4e** (Appendix A); this can probably be attributed to the solvent effect or existence of several emitting states [41,42,43]; lifetimes are on a nanosecond timescale. In each series of compounds **4a**–**c**, **5a**–**c**, and **5d**–**f**, diphenylamino-derivatives **4b**, **5b**, and **5e** are characterized by the highest values of singlet excited-state lifetimes in the range of 1.66–1.85 ns. The values obtained for triazoloquinazolines **4a**–**d** and **5a**–**f** are similar to the lifetimes reported for 4-morpholin-4-yl- and 4-oxoquinazoline systems [31,32]. According to the calculations (Table 2) the relevant radiative decay constants (k_r_) of **4a**, **4b**, and **5a**–**f** are similar and ranges from 52.02 × 10^7^ s^−1^ to 61.25 × 10^7^ s^−1^. In general, energy dissipation in compounds **4a**,**b** and **5a**–**f** mainly occurred through radiative channels due to the high π-conjugation length of molecules (k_r_ > k_nr_), while k_nr_ exceeds k_r_ for derivatives **4c**,**d**,**e**, probably due to considerable twisting of the structure.

We compared the photophysical properties of unsubstituted [1,2,4]triazolo[4,3-*c*]quinazoline fluorophores **4a**–**c** with those of 3-aryl[1,2,4]triazolo[4,3-*c*]quinazolines **A** (Figure 1) reported previously [30]. The region of absorption and the emission band for unsubstituted-at-C(3) position compounds **4a**–**c** is rather similar to their aryl-substituted counterparts **A**; the identical correlation in the influence of aryl fragment nature on photophysical properties is observed. However, removal of the aryl fragment, in general, leads to an increase in the quantum yield in solutions, probably due to the reduction in non-radiative energy losses.

The solid-state luminescent properties of compounds **4a**–**f** and **5a**–**f** were also investigated at room temperature. Triazoloquinazolines show luminescence in the yellow, green, cyan, and blue regions, Figure 5, under irradiation by a hand-held UV lamp.

The measured spectra correlated with visual results, Table 1. The introduction of ethyl group into triazole ring of [1,2,4]triazolo[4,3-*c*]quinazolines **4a**–**f** causes a hypsochromic shift by ~50 nm (for example, compound **4a** regarding **4d**), whereas the emission of **5a**–**f** is not influenced by ethyl substituent. The proximity of the ethyl group to biphenyl moiety, probably, results in twisted conformation of molecule and reduced conjugation length. Some of compounds are characterized by good quantum yields of up to 42%; the values are comparable to 2-(amino[1,1′-biphenyl]-4-yl)-4-(morpholin-4-yl)quinazolines and their 4-oxo counterparts [31,32].

### 2.3. Effects of Solvent Polarity for Compounds ***4*** and ***5***

As long as synthesized triazoloquinazolines **4** and **5** represent push–pull systems with electron-withdrawing triazoloquinazoline core and electron-donating arylamino moiety, separated by a π-system, they are promising fluorosolvatochromic candidates. We studied the absorption and emission properties for some new compounds **4a**, **4d**, **5a**, **5d**, **5e**, and **5f** in the solvents of different polarity (Figure 6 and Appendix A, and Appendix A). The shape and energy of the absorption bands were revealed to be weakly dependent on the solvent polarity, whereas the fluorescence spectra show a strong dependence on the solvent polarity and a remarkable positive solvatochromism (142–193 nm) when going from non-polar cyclohexane to polar MeCN or MeOH. The photograph (Figure 6c) of fluorophore **4a** solutions, as an example, taken under a UV light, exhibited a wide range of colors, from deep blue to orange. The results indicate a low molecular dipole moment in the ground state and the large dipole moment in the excited state. The fluorosolvatochromism suggests a potential intramolecular charge transfer between the donor and acceptor units upon photoexitation. Notably, all compounds show a structured emission in non-polar cyclohexane, and a broad and structureless emission in other solvents of moderate and high polarity suggesting ICT states [44].

To further analyze solvatochromic properties the Lippert–Mataga equation [44,45,46] was employed in which the Stokes shift (Δν) was plotted as a function of the orientation polarizability (Δf) of the solvents, Appendix A. The clear linear trend (R_2_ > 0.92, Table 3) indicates the increase in dipole moment in the excited state compared to the ground state and supports the ICT nature of the excited state. A higher slope for **4d**, **5e**, and **5f** than for other fluorophores suggests that they exhibit a more pronounced charge transfer process.

Onsager radii of the molecules, calculated from the Van der Waals volume [47,48] or by the DFT method, were employed to determine the change in dipole moments Δµ_1_ and Δµ_2,_ respectively, Table 3. The obtained values of the difference Δµ_1_ between the dipole moments of the ground and excited states were calculated to be in the range of 11.05–14.14 D or 33.08–42.81 D. We also calculated Δµ using DFT theory and obtained results ranging from 15.59 to 30.69 D. The underestimation of Δµ_1_ could be attributed to the assuming a spherical model for molecule. Summary of all data suggests remarkable polar structure in excited state. [1,2,4]Triazolo[4,3-*c*]quinazolines **5e** and **5f** exhibit the highest Δµ value in the considered series.

### 2.4. Electrochemical Studies of [1,2,4]Triazoloquinazolines

The electrochemical behavior of the compounds **4a**–**f** and **5a**–**f** was studied using cyclic voltammetry in CH_2_Cl_2_ (Appendix A, Table 4). As can be seen from Appendix A, compounds **4a**–**f** and **5a**–**f** are characterized by quasi-reversible peaks of oxidation, while in the range of the electrochemical stability window of the supporting electrolyte peaks of reduction was not observed. In general, the electrochemical behavior of compounds in the anodic region remains almost unchanged when going from [4,3-*c*] to [1,5-*c*] isomers or from H-substituted to ethyl-substituted derivatives and is determined exclusively by the donor fragment. Based on the obtained oxidation onset potentials, we calculated the HOMO energy for the presented compounds (Table 4). The energies of the HOMO, determined by electrochemistry, match very well with those calculated by DFT.

### 2.5. Quantum-Chemical Calculations

The distribution plots of the HOMOs and LUMOs, as well as energy levels and energy gaps in the gas phase are presented in Appendix A and Table 4. For all the compounds, the HOMO electrons are mainly distributed on the electron-donating aminoaryl unit and phenylene moiety; however, the participation of phenylene spacer is less in carbazol-9-yl-derivatives **4c**,**f** and **5c**,**f** than in its Et_2_N- (**4a**,**d**, **5a**,**d**) and Ph_2_N-containing (**4b**,**e**, **5b**,**e**) counterparts that confirm shorten π-conjugation of the former molecules, due to the twisting of the rigid carbazol-9-yl fragment, and corresponds with the experimental data. The LUMOs plots are similar for the compounds **4a**–**f** and **5a**–**f**; electrons are located in the [1,2,4]triazoloquinazoline framework and the biphenylene part, with partial involvement of the nitrogen atom of donor group in the case of Et_2_N- and Ph_2_N-sabstituted triazoloquinazolines. The value of the energy gap is slightly lower in [1,2,4]triazolo[4,3-*c*]quinazolines **5a**–**f** (E_g_ = 3.47–3.78 eV) than in [4,3-*c*] isomers **4a**–**f** (E_g_ = 3.58–3.95 eV).

Appendix A shows the optimized geometries calculated for the electronic ground state (S_0_) of all molecules in gas phase. The selected dihedral angles α_1_–α_4_, which account for the internal twisting of molecule fragments, are collected in Appendix A. The angles α_1_ present value of 6–7° in triazoloquinazolines **4a**–**c** and value of 10–14° for their Et-substituted counterparts **4d**–**f**, whereas the same angles are close to 1–2° in the case of [1,5-*c*] isomers **5a**–**f**. Moreover, [1,2,4]triazolo[4,3-*c*]quinazolines **4a**–**f** are characterized by highly twisted phenylene residues (angles α_2_ more than 36°) relative to heterocycle core, while α_2_ is around 20° for compounds **5a**–**f**. This difference is most probably the result of the steric hindrance introduced by a hydrogen atom or the Et group of [4,3-*c*]-arranged structure. The angle α_4_ define the deviation from planarity of aminoaryl donor part, the value, predictable, increases when going from Et_2_N- to Ph_2_N- and to carbazol-9-yl derivatives in each of the sets of fluorophores that ascribe to the steric hindrance caused by the phenyl groups of Ph_2_N moiety or the rigid planar structure of the carbazole unit. Overall, triazolo[4,3-*c*]quinazolines are more twisted than their [1,5-*c*] isomers and tend to absorb at higher energetic wavelength displaying a hipsochromically shifted absorption band, that is consistent with the experimental results.

After geometry optimizations, the electronic transition properties (excitation energy (eV), absorption wavelength (nm), oscillator strength (f_osc_), nature of the transition, and major contributions of molecular orbitals) were calculated, Appendix A. The predicted UV/Vis absorption spectra are presented in Appendix A. According to calculations, the lowest excited singlet state (S_1_) for compounds **4**, **5** origins from HOMO/LUMO transitions with contributions >93%, with the energy of Franck–Condon states in the range of 3.13−3.49 eV for compounds of series **4** and 3.09−3.26 eV for compounds of series **5**. The HOMO–LUMO transitions demonstrate a pronounced charge transfer from electron-donating aminoaryl part to [1,2,4]triazoloquinazoline fragment, which is responsible for the underestimation of the energy of the S_1_ transition calculated using TD-DFT by up to 0.3 eV in comparison with the experimental one. In order to further characterize the electronic transitions in the compounds under study, hole–electron analysis for S_0_–S_1_ transitions was carried out (the results are presented in the Table 5) [50].

Calculated parameters indicate that the transitions exhibit notable overlap in the spatial distributions of electrons and holes (S_r_~0.4–0.5), but also significant delocalization, as indicated by high D-indices (more than one bond length) and positive t-indices, meaning that there is a substantial separation of the hole and electron distributions. Based on the results obtained, we can conclude that the S_0_-S_1_ transitions in these compounds have a pronounced charge transfer character, most pronounced for compounds **4c**, **4f**, **5c**, and **5f**.

To gain insights into the fluorescence properties of the compounds **4a**–**f** and **5a**–**f**, the optimized geometries for the electronic excited state (S_1_) were calculated in toluene and MeCN, Appendix A, Figure 7. As can be seen from Appendix A, the deviation of biphenyl residue from plane of triazolo[4,3-*c*]quinazoline core (α_1_) in compounds **4a**–**f** increases more than twice in the excited state compared to the ground state. However, biphenyl moiety tends to shorten angles α_2_ and α_3_. Therefore, triazolo[4,3-*c*]quinazolines **4a**–**f**, characterized by the simultaneous planarization of a biphenyl fragment and the twisting of a polycycle fragment, formed a pincer-like arrangement of the molecule. Contrary, all angles, α_1_–α_3,_ in triazolo[1,5-*c*]quinazolines **5a**–**f** decrease in excited state, forming highly planar 5-biphenyltriazolo[1,5-*c*]quinazoline fragments.

In addition, bond lengths L1 and L2 in **4a**–**f** and **5a**–**f** in the excited states were shortened (Appendix A), indicating conjugation enhancement of structures in excited states conducting a probable ICT process.

## 3. Experimental Methods

### 3.1. General Information

Unless otherwise indicated, all common reagents and solvents were used from commercial suppliers without further purification. Reagent **1** was dried by azeotropic distillation using toluene. Melting points were determined on Boetius-combined heating stages. ^1^H NMR and ^13^C NMR spectra were recorded at room temperature, on a Bruker DRX-400 or Bruker DRX-600 spectrometer (Billerica, MA, USA). Hydrogen chemical shifts (δ in ppm) were referenced to the hydrogen resonance of the corresponding solvent (DMSO-*d*_6_, δ = 2.50 ppm or CDCl_3_, δ = 7.26 ppm). Carbon chemical shifts (δ in ppm) were referenced to the carbon resonances of the solvent (DMSO-*d*_6_, δ = 39.5 ppm CDCl_3_, δ = 77.2 ppm). Peaks are labeled as singlet (s), doublet (d), triplet (t), quartet (q), or multiplet (m). Mass spectra were recorded on the Shimadzu GCMS-QP2010 Ultra instrument (Kyoto, Japan) with electron ionization (EI) of the sample. The elemental analysis was carried out with the use of a Perkin Elmer 2400 Series II C,H,N-analyzer (Waltham, MA, USA).

### 3.2. Photophysical Characterization

UV/vis absorption spectra were recorded on the Shimadzu UV-1800 Spectrophotometer using quartz cells with 1 cm path length at room temperature. Emission spectra were measured on the Horiba FluoroMax-4 (Kyoto, Japan) at room temperature using quartz cells with 1 cm path length. Fluorescence quantum yield of the target compounds in solution and solid state were measured by using the Integrating Sphere Quanta-φ of the Horiba-Fluoromax-4 [41]. Time-resolved fluorescence measurements were carried out using time-correlated single-photon counting (TCSPC) with a nanosecond LED (λ = 370 nm).

### 3.3. Electrochemical Studies

Cyclic voltammetry was carried out on a Metrohm Autolab PGSTAT302N potentiostat (Herisau, Switzerland) with a standard three-electrode configuration. Typically, a three-electrode cell equipped with a platinum disk working electrode (3 mm), a glass carbon disk counter electrode (3 mm), and a Ag/AgNO_3_ (0.01 M) pseudo-reference electrode was used. Measurements were made in dry CH_2_Cl_2_ with tetrabutylammonium hexafluorophosphate (0.1 M) as the supporting electrolyte under an argon atmosphere at a scan rate of 100 mV/s. The potential of reference electrode was calibrated by using the ferrocene/ferrocenium redox couple (Fc^+^/Fc).

### 3.4. Quantum-Chemical Calculations

Conformational search was carried out before DFT calculations using the AQME program [51]. DFT calculations were performed using the Orca 5.0.3 program. The ground-state geometry optimizations were performed at the PBE0-D3BJ/def2-TZVP level of theory in the gas phase. Frequency analyses were carried out at the same theoretical level to ensure that the optimized geometries correspond to a local minimum on the potential energy surface; all compounds were characterized by only real vibrational frequencies. The absorption spectra and optimal geometry of S1-state were calculated by TDDFT at the same theoretical level. The Chemcraft program was used for the visualization [Chemcraft—graphical software for visualization of quantum chemistry computations, Version 3.8, https://www.chemcraftprog.com, accessed on 5 March 2024.

### 3.5. Crystallography

The single crystal of compound **4a** (yellow block of 0.41 × 0.29 × 0.17), **4e** (yellow irregular crystal of 0.44 × 0.26 × 0.15), **5d** (yellow block of 0.48 × 0.39 × 0.27) and **5e** (light yellow block of 0.46 × 0.35 × 0.28) was used for X-ray analysis. Structural studies of the compounds were performed using equipment available in the Collaborative Access Centre “Spectroscopy and Analysis of the Organic compound” at the Postovsky Institute of the Organic Synthesis, Ural Branch, Russian Academy of Sciences. The X-ray diffraction analysis was performed at room temperature on the Xcalibur 3 diffractometer (Oxford Diffraction, Abingdon, UK). Using Olex2 [52], the structure was solved with the ShelXT structure solution program using intrinsic phasing and refined with the ShelXL [53] refinement package using full-matrix least squares minimization. All non-hydrogen atoms were refined in an anisotropic approximation; the H atoms were placed in calculated positions and refined isotropically in the “rider” model.

Crystal data for **4a** C_25_H_23_N_5_ includes the following: M = 393.48, monoclinic, a = 7.4970(4) Å, b = 12.6068(6) Å, c = 22.0871(14) Å, α = 90°, β = 96.507(5)°, γ = 90°, V = 2074.07(19) Å^3^, space group P2_1_/n, Z = 4, and μ(Mo Kα) = 0.077 mm^−1^. On the angles 2.46 < 2Θ < 30.91°, 9749 reflections were measured and 2557 unique (R_int_ = 0.0739), which were used in all calculations. Goodness to fit was recorded at F^2^ 0.996; the final R_1_ = 0.1548, wR_2_ = 0.1555 (all data) and R_1_ = 0.0601, wR_2_ = 0.1139 (I > 2σ(I)). The largest diff. peak and hole was 0.135 and –0.187 ēÅ^−3^.

Crystal data for **4e** C_35_H_27_N_5_ includes the following: M = 517.61, monoclinic, a = 11.2955(12) Å, b = 13.9226(11) Å, c = 17.6642(17) Å, α = 90°, β = 103.110(10)°, γ = 90°, V = 2705.5(5) Å^3^, space group P2_1_/n, Z = 4, and μ(Mo Kα) = 0.076 mm^−1^. On the angles 3.522 < 2Θ < 26.367°, 19,711 reflections measured, 2529 unique (R_int_ = 0.0847), which were used in all calculations. Goodness to fit was recorded at F^2^ 0.985; the final R_1_ = 0.1503, wR_2_ = 0.2332 (all data) and R_1_ = 0.0672, wR_2_ = 0.1692 (I > 2σ(I)). The largest diff. peak and hole was 0.297 and –0.159 ēÅ^−3^.

Crystal data for **5d** C_27_H_27_N_5_ includes the following: M = 421.53, triclinic, a = 8.7546(7) Å, b = 11.8118(11) Å, c = 12.9418(10) Å, α = 111.946(7)°, β = 100.965(7)°, γ = 106.434(7)°, V = 1122.79(17) Å^3^, space group P −1, Z = 2, and μ(Mo Kα) = 0.076 mm^−1^. On the angles 3.597 < 2Θ < 31.000°, 11,319 reflections measured, 3163 unique (R_int_ = 0.0487) which were used in all calculations. Goodness to fit was recorded at F^2^ 1.022; the final R_1_ = 0.1244, wR_2_ = 0.2334 (all data) and R_1_ = 0.0672, wR_2_ = 0.1711 (I > 2σ(I)). The largest diff. peak and hole was 0.292 and –0.204 ēÅ^−3^.

Crystal data for **5e** C_35_H_27_N_5_ includes the following: M = 517.61, monoclinic, a = 9.4980(7) Å, b = 29.6579(18) Å, c = 10.0462(10) Å, α = 90°, β = 107.435(7)°, γ = 90°, V = 2699.9(3) Å^3^, space group P2_1_/n, Z = 4, and μ(Mo Kα) = 0.077 mm^−1^. On the angles 3.550 < 2Θ < 29.570°, 19,422 reflections measured, 3419 unique (R_int_ = 0.1029) which were used in all calculations. Goodness to fit was recorded at F^2^ 0.957; the final R_1_ = 0.1511, wR_2_ = 0.2311 (all data) and R_1_ = 0.0780, wR_2_ = 0.1758 (I > 2σ(I)). The largest diff. peak and hole was 0.236 and –0.286 ēÅ^−3^.

The results of X-ray diffraction analysis for compounds **4a**, **4e**, **5d**, and **5e** were deposited in the Cambridge Crystallographic Data Centre (CCDC 2,336,242 for **4a**, CCDC 2,336,250 for **4e**, CCDC 2,336,251 for **5d**, and CCDC 2,336,243 for **5e**). The data are free and can be available at www.ccdc.cam.ac.uk.

### 3.6. Synthesis of Compounds ***2a**,**b***, ***3a**,**b***, ***4a**–**f*** and ***5a**–**f***

#### 3.6.1. General Procedure for the Synthesis of [1,2,4]Triazolo[4,3-*c*]quinazolines (**2a**,**b**)

Method 1. In a round-bottom flask equipped with a magnetic stirred bar, 2-(4-bromophenyl)-4-hydrazinoquinazoline **1** (0.23 g, 0.72 mmol) in absolute ethanol (17 mL) and corresponding ortho ester (4.30 mmol) were added. The mixture was refluxed for 4 h. Condenser was equipped with a calcium chloride drying tube. After cooling down and partial evaporation the solid was filtered off, washed with water, dried and used in the next step without further purification. A pure sample for analysis was obtained by crystallization from DMSO.

Method 2. In a round-bottom flask equipped with a magnetic stirred bar, dried 2-(4-bromophenyl)-4-hydrazinoquinazoline **1** (0.28 g, 0.89 mmol) and corresponding ortho ester (7.20 mmol) were added. The mixture was refluxed for 4 h. A condenser was equipped with a calcium chloride drying tube. After cooling down the solid was filtered off, washed with water and dried.

*5-(4-Bromophenyl)-[1,2,4]triazolo[4,3-c]quinazoline* (**2a**). The general procedure was applied using **1** and triethyl orthoformate: colorless powder, yield 83% (method 1), yield 86% (method 2); mp 250–252 °C; ^1^H NMR (DMSO-*d*_6_, 400 MHz) δ 7.78–7.86 (4H, m), 7.99–8.01 (3H, m), 8.52–8.54 (1H, m, H-7 or H-10), 9.32 (1H, s, H-3); ^13^C {^1^H} NMR (DMSO-*d*_6_, 100 MHz, 40 °C) δ 115.5, 122.7, 125.1, 128.2, 129.1, 130.9, 131.4, 131.8, 131.9, 137.0, 140.8, 144.2, 147.4; EIMS *m*/*z* 326 [M + 2]^+^ (96), 325 [M + 1]^+^ (54), 324 [M]^+^ (100), 298 [M − N_2_ + 2]^+^ (16), 217 [M − N_2_ − Br]^+^ (37); C_15_H_9_BrN_4_ (324.00).

*5-(4-Bromophenyl)-3-ethyl-[1,2,4]triazolo[4,3-c]quinazoline* (**2b**). The general procedure was applied using **1** and triethyl orthopropionate: beige powder, yield 83%; mp 189–191 °C; ^1^H NMR (DMSO-*d*_6_, 400 MHz) δ 1.12 (3H, t, ^3^*J* = 7.3 Hz, CH_3_), 2.30 (2H, q, ^3^*J* = 7.3 Hz, CH_2_), 7.75–7.85 (6H, m), 7.92–7.94 (1H, m), 8.48–8.50 (1H, m, H-7 or H-10); ^13^C {^1^H} NMR (DMSO-*d*_6_, 100 MHz) δ 11.2 (CH_3_), 21.1 (CH_2_), 116.3, 122.4, 124.1, 127.9, 129.2, 131.1, 131.2, 132.7, 140.2, 145.0, 148.5, 159.7; EIMS *m*/*z* 354 [M + 2]^+^ (97), 353 [M + 1]^+^ (83), 352 [M]^+^ (100), 337 [M − CH_3_]^+^ (19), 102 (74); C_17_H_13_BrN_4_ (352.03).

#### 3.6.2. General Procedure for the Synthesis of [1,2,4]Triazolo[1,5-*c*]quinazolines (**3a**,**b**)

Method 1. In a round-bottom flask equipped with a magnetic stirred bar, corresponding [1,2,4]triazolo[4,3-*c*]quinazoline **2** (0.61 mmol) and glacial acetic acid (5 mL) were added together. The mixture was refluxed for 4 h. After cooling down the water was added until the formation of precipitate. The product was filtered off and washed with water, dried and used in the next step without further purification. A pure sample for analysis was obtained by crystallization from DMSO.

Method 2. In a round-bottom flask equipped with a magnetic stirred bar, 2-(4-bromophenyl)-4-hydrazino-quinazoline **1** (0.95 mmol) in glacial acetic acid (7 mL) and a corresponding orthoester (4.70 mmol) were added together. The mixture was refluxed for 16 h. After cooling down the water was added until the formation of precipitate. The product was filtered off and washed with water, dried and used in the next step without further purification. Pure sample for analysis was obtained by crystallization from DMSO.

*5-(4-Bromophenyl)-[1,2,4]triazolo[1,5-c]quinazoline* (**3a**). The general procedure was applied using [1,2,4]triazolo[4,3-*c*]quinazoline **2a** as the starting material: beige powder, yield 93% (method 1), yield 91% (method 2); mp 186–188 °C; ^1^H NMR (DMSO-*d*_6_, 400 MHz) δ 7.79–7.84 (3H, m, H-3′, H-5′, H-8 or H-9), 7.93–7.96 (1H, m, H-8 or H-9), 8.10–8.12 (1H, m, H-7 or H-10), 8.50–8.54 (3H, m, H-2′, H-6′, H-7 or H-10), 8.63 (1H, s, H-2); ^13^C {^1^H} NMR (DMSO-*d*_6_, 100 MHz, 50 °C) δ 117.0, 123.0, 125.2, 128.2, 128.7, 130.4, 131.2, 132.0, 132.2, 141.9, 144.9, 151.2, 153.5; EIMS *m*/*z* 326 [M + 2]^+^ (100), 325 [M + 1]^+^ (48), 324 [M]^+^ (97), 298 [M − N_2_ + 2]^+^ (17), 217 [M − N_2_ − Br]^+^ (49); C_15_H_9_BrN_4_ (324.00).

*5-(4-Bromophenyl)-2-ethyl-[1,2,4]triazolo[1,5-c]quinazoline* (**3b**). The general procedure was applied using [1,2,4]triazolo[4,3-c]quinazoline **2b** as the starting material: colorless powder, yield 93% (method 1), yield 89% (method 2); mp 132–134 °C; ^1^H NMR (DMSO-*d*_6_, 400 MHz) δ 1.39 (3H, t, ^3^*J* = 7.5 Hz, CH_3_), 2.97 (2H, q, ^3^*J* = 7.5 Hz, CH_2_), 7.80–7.83 (1H, m, H-8 or H-9), 7.86–7.88 (2H, m, H-3′, H-5′), 7.93–7.97 (1H, m, H-8 or H-9), 8.10–8.12 (1H, m, H-7 or H-10), 8.85–8.47 (3H, m, H-2′, H-6′, H-7 or H-10); ^13^C {^1^H} NMR (DMSO-*d*_6_, 100 MHz, 50 °C) δ 12.1 (CH_3_), 21.5 (CH_2_), 116.7, 123.1, 125.2, 128.3, 128.6, 130.7, 131.2, 132.0, 132.2, 142.0, 144.7, 151.7, 167.6; EIMS *m*/*z* 354 [M + 2]^+^ (68), 353 [M + 1]^+^ (47), 352 [M]^+^ (70), 339 [M − CH_3_ + 2]^+^ (14), 102 (100); C_17_H_13_BrN_4_ (352.03).

*4-Bromo-N-(2-(3-ethyl-1H-[1,2,4]triazol-5-yl)phenyl)benzamide* (**3b′**). The method 2 was applied. The reaction was stopped after 8 h. After cooling down the water was added until the formation of a precipitate. The product was filtered off, washed with water, and dried. Then it was purified by column chromatograpgy on silica gel, eluent EtOAc/hexane (3:7) to pure EtOAc. Pale orange powder, mp 220–222 °C; ^1^H NMR (DMSO-*d*_6_, 400 MHz) δ 1.33 (3H, t, ^3^*J* = 7.4 Hz, CH_3_), 2.88 (2H, q, ^3^*J* = 7.4 Hz, CH_2_), 7.20–7.24 (1H, m, benzo), 7.44–7.46 (1H, m, benzo), 7.80–7.82 (2H, m, 4-BrC_6_H_4_), 8.01–8.04 (2H, m, 4-BrC_6_H_4_), 8.17–8.19 (1H, m, benzo), 8.74–8.76 (1H, m, benzo), 12.8 (1H, s, NH), 14.2 (1H, s, NH); EIMS *m*/*z* 372 [M + 2]^+^ (75), 370 [M]^+^ (78), 185 [C_7_H_4_BrO + 2]^+^ (98), 183 [C_7_H_4_BrO]^+^ (100); C_17_H_15_BrN_4_O (370.04).

#### 3.6.3. General Procedures for the Synthesis of Target Products **4a**–**f** and **5a**–**f**

The corresponding boronic acid or boronic acid pinacol ester (0.64 mmol), PdCl_2_(PPh_3_)_2_ (48 mg, 68 μmol), PPh_3_ (36 mg, 136 μmol), saturated solution of K_2_CO_3_ (3.7 mL) and EtOH (3.7 mL) were added to the suspension of the corresponding derivative **2a**,**b** or **3a**,**b** (0.60 mmol) in toluene (22 mL). The mixture was stirred at 85 °C for 12–14 h in argon atmosphere in round-bottom pressure flask equipped with magnetic stirred bar. The reaction mixture was cooled to room temperature, and EtOAc/H_2_O (10/10 mL) mixture was added. The organic layer was separated, additionally washed with water (10 mL), and evaporated at reduced pressure. The product was purified by column chromatography on silica gel, hexane/ethyl acetate mixture was used as an eluent.

*5-(4′-Diethylamino-[1,1′]-biphenyl-4-yl)-[1,2,4]triazolo[4,3-c]quinazoline* (**4a**). The general procedure was applied using [1,2,4]triazolo[4,3-*c*]quinazoline **2a** and 4-(diethylamino)phenylboronic acid. Eluent for column chromatography: EtOAc/hexane (2/8) → EtOAc/hexane (1/1). Yellow powder, yield 50%; mp 155–157 °C; ^1^H NMR (CDCl_3_, 400 MHz) δ 1.22 (6H, t, ^3^*J* = 6.9 Hz, 2CH_3_), 3.43 (4H, q, ^3^*J* = 6.9 Hz, 2CH_2_), 6.78–8.80 (2H, m, 2CH_phenylene_), 7.58–7.60 (2H, m, 2CH_phenylene_), 7.71–7.73 (1H, m, H-8 or H-9), 7.78–7.82 (3H, m, 2CH_phenylene_, H-8 or H-9), 7.96–7.98 (2H, m, 2CH_phenylene_), 8.88–8.68 (1H, m, H-7 or H-10), 9.07 (1H, s, H-3); ^13^C {^1^H} NMR (CDCl_3_, 100 MHz) δ 12.7 (2CH_3_), 44.5 (2CH_2_), 112.0, 115.9, 123.6, 125.8, 126.6, 128.2, 128.7, 129.0, 129.1, 132.0, 136.3, 141.7, 144.8, 145.0, 148.1, 148.7; EIMS *m*/*z* 394 [M + 1]^+^ (20), 393 [M]^+^ (67), 378 [M − CH_3_]^+^ (100); anal. calcd for C_25_H_23_N_5_ (393.20): C 76.31, H 5.89, N 17.80%. Found C 76.25, H 6.11, N 17.56%.

*5-(4′-Diphenylamino-[1,1′]-biphenyl-4-yl)-[1,2,4]triazolo[4,3-c]quinazoline* (**4b**). The general procedure was applied using [1,2,4]triazolo[4,3-*c*]quinazoline **2a** and 4-(diphenylamino)phenylboronic acid. Eluent for column chromatography: EtOAc/hexane (1/3) → EtOAc. Additionally, the product was recrystallized from DMSO. Yellow powder, yield 62%; mp 110–112 °C; ^1^H NMR (CDCl_3_, 400 MHz) δ 7.06–7.10 (2H, m, 2CH_phenyl_), 7.16–7.19 (6H, m, 4CH_phenyl,_ 2CH_phenylene_), 7.29–7.32 (4H, m, 4CH_phenyl_), 7.55–7.57 (2H, m, 2CH_phenylene_), 7.73–7.76 (1H, m, H-8 or H-9), 7.81–7.85 (3H, m, 2CH_phenylene_, H-8 or H-9), 8.00–8.07 (3H, m, 2CH_phenylene,_ H-7 or H-10), 8.68–8.70 (1H, m, H-7 or H-10), 9.06 (1H, s, H-3); ^13^C {^1^H} NMR (CDCl_3_, 150 MHz) 115.9, 123.4, 123.6, 123.7, 125.0, 127.5, 128.0, 128.8, 129.1, 129.4, 129.6, 130.5, 132.2, 132.8, 136.2, 141.7, 144.5, 144.6, 147.5, 148.5, 148.8; EIMS *m*/*z* 490 [M + 1]^+^ (36), 489 [M]^+^ (100); anal. calcd for C_33_H_23_N_5_×DMSO×1/2H_2_O: C 72.24, H 4.78, N 12.76%. Found C 72.38, H 4.02, N 12.74%.

*5-(4′-(9H-Carbazol-9-yl)-[1,1′]-biphenyl-4-yl)-[1,2,4]triazolo[4,3-c]quinazoline* (**4c**). The general procedure was applied using [1,2,4]triazolo[4,3-*c*]quinazoline **2a** and 4-(9*H*-carbazol-9-yl)phenylboronic acid pinacol ester. Eluent for column chromatography: EtOAc/hexane (1/3) → EtOAc. Additionally, the product was recrystallized from DMSO. Beige powder, yield 63%; mp > 250 °C; ^1^H NMR (CDCl_3_, 400 MHz) δ 7.31–7.34 (2H, m, 2CH_carbaz_.), 7.43–7.47 (2H, m, 2CH_carbaz_.), 7.50–7.52 (2H, m, 2CH_carbaz_.), 7.73–7.78 (3H, m), 7.83–7.87 (1H, m, H-8 or H-9), 8.92–8.94 (2H, m), 8.97–8.99 (2H, m), 8.08–8.12 (3H, m) 8.17–8.19 (2H, m), 8.70–8.72 (1H, m, H-7 or H-10), 9.09 (1H, s, H-2); ^13^C {^1^H} NMR (CDCl_3,_ 150 MHz) δ 109.9, 116.0, 120.4, 120.6, 123.7, 123.8, 126.2, 127.7, 128.2, 128.9, 129.3, 129.5, 131.5, 132.3, 136.2, 138.2, 138.6, 140.8, 141.6, 144.1, 144.4, 148.8; EIMS *m*/*z* 488 [M + 1]^+^ (38), 487 [M]^+^ (100); anal. calcd for C_33_H_21_N_5_ (487.18): C 81.29, H 4.34, N 14.34%. Found C 81.35, H 4.18, N 14.67%.

*5-(4′-Diethylamino-[1,1′]-biphenyl-4-yl)-2-ethyl-[1,2,4]triazolo[4,3-c]quinazoline* (**4d**). The general procedure was applied using [1,2,4]triazolo[4,3-*c*]quinazoline **2b** and 4-(diethylamino)phenylboronic acid. Eluent for column chromatography: EtOAc/hexane (1/3) → EtOAc. Additionally, the product was recrystallized twice from the mixture of EtOAc/hexane. Pale yellow powder, yield 61%; mp 154–156 °C; ^1^H NMR (CDCl_3_, 400 MHz) δ 1.19–1.24 (9H, m, 3CH_3_), 2.57 (2H, q, ^3^*J* = 7.3 Hz, CH_2_), 3.43 (4H, q, ^3^*J* = 7.3 Hz, 2CH_2_), 6.78–6.80 (2H, m, 2CH_phenylene_), 7.59–7.64 (4H, m, 4CH_phenylene_), 7.64–7.69 (1H, m, H-8 or H-9), 7.75–7.77 (3H, m, 2CH_phenylene_, H-8 or H-9), 7.96–7.8 (1H, m, H-7 or H-10), 8.66–8.68 (1H, m, H-7 or H-10); ^13^C {^1^H} NMR (CDCl_3_, 100 MHz) δ 11.9 (CH_3_), 12.2 (2CH_3_), 21.9 (CH_2_), 44.5 (2CH_2_), 112.0, 116.6, 123.3, 125.8, 125.9, 128.0, 128.2, 128.9, 129.0, 130.4, 131.5, 140.8, 143.9, 145.9, 147.9, 149.6, 150.5; EIMS *m*/*z* 422 [M + 1]^+^ (25), 421 [M]^+^ (76), 406 [M − CH_3_]^+^ (100); anal. calcd for C_27_H_27_N_5_ (421.23): C 76.93, H 6.46, N 16.61%. Found C 76.73, H 6.24, N 16.31%.

*5-(4′-Diphenylamino-[1,1′]-biphenyl-4-yl)-2-ethyl-[1,2,4]triazolo[4,3-c]quinazoline* (**4e**). The general procedure was applied using [1,2,4]triazolo[4,3-*c*]quinazoline **2b** and 4-(diphenylamino)phenylboronic acid. Eluent for column chromatography: EtOAc/hexane (7/3) → EtOAc/hexane (1/1). Pale yellow powder, yield 72%; mp 154–156 °C; ^1^H NMR (CDCl_3_, 400 MHz) δ 1.22 (3H, t, *^3^J* = 7.2 Hz, CH_3_), 2.55 (2H, q, ^3^*J* = 7.2 Hz, CH_2_), 7.06–7.09 (2H, m, 2CH_phenyl_), 7.15–7.20 (6H, m, 4CH_phenyl_, 2CH_phenylene_), 7.28–7.32 (4H, m, 4CH_phenyl_), 7.56–7.58 (2H, m, 2CH_phenylene_), 7.66–7.73 (3H, m, 2CH_phenylene_, H-8 or H-9), 7.76–7.80 (3H, m, 2CH_phenylene_, H-8 or H-9), 7.97–7.99 (1H, m, H-7 or H-10), 8.67–8.69 (1H, m, H-7 or H-10); ^13^C {^1^H} NMR (CDCl_3_, 150 MHz) δ 12.0 (CH_3_), 22.0 (CH_2_), 116.7, 123.4, 123.5, 123.6, 124.9, 126.8, 128.0, 128.3, 129.2, 129.3, 129.5, 131.6, 131.8, 133.1, 140.9, 143.4, 145.7, 147.6, 148.3, 149.6, 150.5; EIMS *m*/*z* 518 [M + 1]^+^ (44), 517 [M]^+^ (100); anal. calcd for C_35_H_27_N_5_ (517.23): C 81.21, H 5.26, N 13.53%. Found C 81.05, H 5.11, N 13.22%.

*5-(4′-(9H-Carbazol-9-yl)-[1,1′]-biphenyl-4-yl)-2-ethyl-[1,2,4]triazolo[4,3-c]quinazoline* (**4f**). The general procedure was applied using [1,2,4]triazolo[4,3-*c*]quinazoline **2b** and 4-(9*H*-carbazol-9-yl)phenylboronic acid pinacol ester. Eluent for column chromatography: EtOAc/hexane (1/3) → EtOAc. Additionally, the product was washed with hexane. Pale beige powder, yield 72%; mp 255–257 °C; ^1^H NMR (DCCl_3_, 400 MHz) δ 1.26 (3H, t, *^3^J* = 7.3 Hz, CH_3_), 2.59 (2H, q, ^3^*J* = 7.3 Hz, CH_2_), δ 7.31–7.34 (2H, m, 2CH_carbaz_.), 7.43–7.47 (2H, m, 2CH_carbaz_.), 7.50–7.52 (2H, m, 2CH_carbaz_.), 7.71–7.81 (6H, m), 7.92–7.94 (4H, m), 7.99–8.01 (1H, m, H-7 or H-10), 8.17–8.19 (2H, m), 8.69–8.71 (1H, m, H-7 or H-10); ^13^C {^1^H} (CDCl_3_, 100 MHz) δ 12.0 (CH_3_), 22.1 (CH_2_), 109.9, 116.8, 120.3, 120.6, 123.5, 123.7, 126.2, 127.5, 127.7, 128.4, 128.8, 129.5, 131.8, 132.7, 138.8, 140.8, 140.9, 143.1, 145.4, 149.7, 150.4; EIMS *m*/*z* 516 [M + 1]^+^ (42), 515 [M]^+^ (100); anal. calcd for C_35_H_25_N_5_ (515.21): C 81.51, H 4.89, N 13.58%. Found C 80.43, H 5.20, N 13.26%.

*5-(4′-Diethylamino-[1,1′]-biphenyl-4-yl)-[1,2,4]triazolo[1,5-c]quinazoline* (**5a**). The general procedure was applied using [1,2,4]triazolo[1,5-*c*]quinazoline **3a** and 4-(diethylamino)phenylboronic acid. Eluent for column chromatography: EtOAc/hexane (1/9). Yellow powder, yield 77%; mp 170–172 °C; ^1^H NMR (CDCl_3_, 400 MHz) δ 1.22 (6H, t, *^3^J* = 7.0 Hz, 2CH_3_), 3.42 (4H, q, ^3^*J* = 7.0 Hz, 2CH_2_), 6.77–6.79 (2H, m, 2CH_phenylene_), 7.69–7.73 (1H, m, H-8 or H-9), 7.77–7.80 (2H, m, 2CH_phenylene_), 7.83–7.87 (1H, m, H-8 or H-9), 8.12–8.14 (1H, m, H-7 or H-10), 8.48 (1H, s, H-2), 8.55–8.61 (3H, m, 2CH_phenylene,_ H-7 or H-10); ^13^C {^1^H} NMR (CDCl_3_, 100 MHz) δ 13.1 (2CH_3_), 44.9 (2CH_2_), 112.3, 117.8, 124.0, 126.1, 126.9, 128.5, 128.6, 129.1, 129.2, 131.2, 132.6, 143.5, 145.0, 147.0, 148.2, 152.5, 153.9; EIMS *m*/*z* 394 [M + 1]^+^ (21), 393 [M]^+^ (70), 378 [M − CH_3_]^+^ (100); anal. calcd for C_25_H_23_N_5_ (393.20): C 76.31, H 5.89, N 17.80%. Found C 76.55, H 6.26, N 18.21%.

*5-(4′-Diphenylamino-[1,1′]-biphenyl-4-yl)-[1,2,4]triazolo[1,5-c]quinazoline* (**5b**). The general procedure was applied using [1,2,4]triazolo[1,5-*c*]quinazoline **3a** and 4-(diphenylamino)phenylboronic acid. Eluent for column chromatography: EtOAc/hexane (3/17). Yellow-green powder, yield 36%; mp 170–172 °C; ^1^H NMR (CDCl_3_, 400 MHz) δ 7.05–7.09 (2H, m, 2CH_phenyl_), 7.16–7.18 (6H, m, 4CH_phenyl,_ 2CH_phenylene_), 7.26–7.32 (4H, m, 4CH_phenyl_), 7.57–7.59 (2H, m, 2CH_phenylene_), 7.72–7.75 (1H, m, H-8 or H-9), 7.72–7.75 (2H, m, 2CH_phenylene_), 7.85–7.89 (1H, m, H-8 or H-9), 8.14–8.16 (1H, m, H-7 or H-10), 8.49 (1H, s, H-2), 8.57–8.59 (1H, m, H-7 or H-10), 8.62–8.64 (2H, m, 2CH_phenylene_); ^13^C {^1^H} NMR (100 MHz, CDCl_3_) δ 117.6, 123.4, 123.6, 124.9, 126.6, 128.1, 128.5, 128.9, 129.5, 130.0, 131.0, 132.4, 133.7, 143.1, 144.0, 146.5, 147.6, 148.2, 152.2, 153.6; EIMS *m*/*z* 490 [M + 1]^+^ (40), 489 [M]^+^ (100); anal. calcd for C_33_H_23_N_5_ (489.20): C 80.96, H 4.74, N 14.31%. Found C 80.88, H 5.00, N 14.04%.

*5-(4′-(9H-Carbazol-9-yl)-[1,1′]-biphenyl-4-yl)-[1,2,4]triazolo[1,5-c]quinazoline* (**5c**). The general procedure was applied using [1,2,4]triazolo[1,5-*c*]quinazoline **3a** and 4-(9*H*-carbazol-9-yl)phenylboronic acid pinacol ester. After cooling the reaction mixture product was filtered off, washed with hexane. Pale grey powder, yield 67%; mp 257–259 °C; ^1^H NMR (DMSO-*d*_6_, 600 MHz) δ 7.31–7.34 (2H, m, 2CH_carbaz_.), 7.46–7.51 (4H, m, 4CH_carbaz_.), 7.80–7.81 (2H, m), 7.85–7.87 (1H, m, H-8 or H-9), 7.99–8.01 (1H, m, H-8 or H-9), 8.10–8.11 (2H, m), 8.13–8.14 (2H, m), 8.18–8.19 (1H, m, H-7 or H-10), 8,27–8.28 (2H, m), 8.53–8.54 (1H, m, H-7 or H-10), 8.69–8.71 (2H, m, 2CH_phenylene_), 8.79 (1H, s, H-2); ^13^C {^1^H} NMR (150 MHz, DMSO-*d*_6,_ 55 °C) δ 109.5, 117.1, 120.1, 120.5, 122.8, 123.2, 126.3, 126.5, 127.1, 128.4, 128.6, 128.7, 130.6, 131.0, 132.4, 136.9, 138.0, 140.0, 142.1, 142.2, 145.7, 151.4, 153.6; EIMS *m*/*z* 488 [M + 1]^+^ (37), 487 [M]^+^ (100); anal. calcd for C_33_H_21_N_5_ (487.20): C 81.29, H 4.34, N 14.34%. Found C 81.18, H 4.15, N 14.39%.

*5-(4′-Diethylamino-[1,1′]-biphenyl-4-yl)-2-ethyl-[1,2,4]triazolo[1,5-c]quinazoline* (**5d**). The general procedure was applied using [1,2,4]triazolo[1,5-*c*]quinazoline **3b** and 4-(diethylamino)phenylboronic acid. Eluent for column chromatography: EtOAc/hexane (1/2) → EtOAc/hexane (1/1). Additionally, the product was recrystallized from a mixture of CH_2_Cl_2_/hexane. Yellow powder, yield 51%; mp 116–118 °C; ^1^H NMR (CDCl_3_, 400 MHz) δ 1.22 (6H, t, *^3^J* = 6.5 Hz, 2CH_3_), 1.51 (3H, t, *^3^J* = 7.5 Hz, CH_3_), 3.08 (2H, q, ^3^*J* = 7.5 Hz, CH_2_), 3.43 (4H, q, ^3^*J* = 6.5 Hz, 2CH_2_), 6.78–6.80 (2H, m, 2CH_phenylene_), 7.60–7.62 (2H, m, 2CH_phenylene_), 7.66–7.70 (1H, m, H-8 or H-9), 7.77–7.85 (3H, m, 2CH_phenylene_, H-8 or H-9), 8.09–8.11 (1H, m, H-7 or H-10), 8.53–8.55 (1H, m, H-7 or H-10), 8.62–8.64 (2H, m, 2CH_phenylene_); ^13^C {^1^H} NMR (CDCl_3_, 100 MHz) δ 12.8 (2CH_3_), 12.9 (CH_3_), 22.6 (CH_2_), 44.6 (2CH_2_), 112.0, 117.2, 123.7, 125.8, 126.8, 128.0, 128.3, 128.8, 129.1, 130.9, 132.0, 143.2, 144.5, 146.6, 147.9, 152.7, 168.5; EIMS *m*/*z* 422 [M + 1]^+^ (25), 421 [M]^+^ (80), 406 [M − CH_3_]^+^ (100); anal. calcd for C_27_H_27_N_5_ (421.23): C 76.93, H 6.46, N 16.61%. Found C 76.72, H 6.22, N 16.42%.

*5-(4′-Diphenylamino-[1,1′]-biphenyl-4-yl)-2-ethyl-[1,2,4]triazolo[1,5-c]quinazoline* (**5e**). The general procedure was applied using [1,2,4]triazolo[1,5-*c*]quinazoline **3b** and 4-(diphenylamino)phenylboronic acid. Eluent for column chromatography: hexane → EtOAc/hexane (8/2). Additionally, the product was washed with hexane. Yellow–green powder, yield 69%; mp 185–187 °C; ^1^H NMR (CDCl_3_, 400 MHz) δ 1.51 (3H, t, *^3^J* = 7.2 Hz, CH_3_), 3.08 (2H, q, ^3^*J* = 7.2 Hz, CH_2_), 7.05–7.08 (2H, m, 2CH_phenyl_), 7.16–7.18 (6H, m, 4CH_phenyl,_ 2CH_phenylene_), 7.28–7.31 (4H, m, 4CH_phenyl_), 7.57–7.59 (2H, m, 2CH_phenylene_), 7.68–7.71 (1H, m, H-8 or H-9), 7.79–7.86 (3H, m, 2CH_phenylene_, H-8 or H-9), 8.10–8.12 (1H, m, H-7 or H-10), 8.54–8.56 (1H, m, H-7 or H-10), 8.65–8.67 (2H, m, 2CH_phenylene_); ^13^C {^1^H} NMR (100 MHz, CDCl_3_) δ 12.8 (CH_3_), 22.5 (CH_2_), 117.3, 123.4, 123.6, 123.7, 124.9, 126.5, 128.0, 128.2, 128.8, 129.5, 130.2, 131.0, 132.0, 133.8, 143.1, 143.8, 146.2, 147.6, 148.1, 152.7, 168.6; EIMS *m*/*z* 518 [M + 1]^+^ (41), 517 [M]^+^ (100); anal. calcd for C_35_H_27_N_5_ (517.23): C 81.21, H 5.26, N 13.53%. Found C 82.34, H 5.48, N 14.04%.

*5-(4′-(9H-Carbazol-9-yl)-[1,1′]-biphenyl-4-yl)-2-ethyl-[1,2,4]triazolo[1,5-c]quinazoline* (**5f**). The general procedure was applied using [1,2,4]triazolo[1,5-c]quinazoline **3b** and 4-(9*H*-carbazol-9-yl)phenylboronic acid pinacol ester. After cooling the reaction mixture product was filtered off, washed with hexane and recrystallized from DMSO. Pale beige powder, yield 55%; mp 286–288 °C; ^1^H NMR (DMSO-*d*_6_, 400 MHz) δ 1.49 (3H, t, *^3^J* = 7.5 Hz, CH_3_), 3.03 (2H, q, ^3^*J* = 7.5 Hz, CH_2_), δ 7.26–7.30 (2H, m, 2CH_carbaz_.), 7.41–7.44 (2H, m, 2CH_carbaz_.), 7.49–7.51 (2H, m, 2CH_carbaz_.), 7.76–7.80 (3H, m), 7.90–7.94 (1H, m, H-8 or H-9), 8.00–8.02 (2H, m), 8.08–8.12 (3H, m), 8.18–8.19 (2H, m), 8.48–8.50 (1H, m, H-7 or H-10), 8.77–8.79 (2H, m, 2CH_phenylene_); ^13^C NMR was not recorded due to poor solubility of the sample. EIMS *m*/*z* 516 [M + 1]^+^ (40), 515 [M]^+^ (100); anal. calcd for C_35_H_25_N_5_ (515.21): C 81.51, H 4.89, N 13.58%. Found C 81.46, H 5.04, N 14.05%.

## 4. Conclusions

We have developed a synthetic approach to 5-(4-bromophenyl)-[1,2,4]triazolo[4,3-*c*]quinazolines and their [1,5-*c*]-isomers. The bromophenyl derivatives were successfully functionalized by introducing aminoaryl donor fragments via palladium-catalyzed cross-coupling reactions with boronic acids or their esters; two series of 5-(4′-EDG-[1,1′]-biphenyl-4-yl)[1,2,4]triazoloquinazoline fluorophores were prepared. The structures of the ring-opening product and target fluorophores were unambiguously confirmed by NMR spectroscopy, mass-spectrometry, and XRD data. The photophysical properties of [1,2,4]triazoloquinazolines have been studied in two solvents and in the solid state. 9*H*-Carbazol-9-yl-containing triazoloquinazolines are characterized by absorption maxima in the range of 312–328 nm in MeCN, whereas the band of their Et_2_N or Ph_2_N counterparts is red-shifted, and the maxima are located in the 340–375 nm range; the absorption band slightly shifts to red region in toluene compared to the MeCN solution. All the compounds are emissive in the blue–cyan region in toluene and in the yellow–orange region in MeCN, with different fluorescence intensities and quantum yields. Some of triazolo[4,3-*c*]quinazolines exhibited medium to high quantum yields, both in solution and in solid state. Triazoloquinazolines with a [1,5-*c*] annelation type turned out to be more effective fluorophores with quantum yields of over 75% in toluene solutions. The presence of the ethyl group has considerable impact on photophysical properties of triazolo[4,3-*c*]quinazolines due to steric hindrance between the close-located biphenyl substituent and ethyl group. The quantum yields of presented compounds are found to be higher than those of their 3-aryl-[1,2,4]triazolo[4,3-*c*]quinazoline counterparts. The experimental findings were conducted by theoretical calculations. Notably, synthesized push–pull organic systems exhibit strong fluorosolvatochromism as a consequence of the large dipole moment in the excited state, with the strong emission intensity in aprotic solvents, thus making them interesting candidates for practical application as fluorescence probes.

## Data Availability

The data are available on request from the corresponding authors.

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
