# Peer review of "Design, Synthesis, and Photophysical Properties of 5-Aminobiphenyl Substituted [1,2,4]Triazolo[4,3-c]- and [1,2,4]Triazolo[1,5-c]quinazolines"

_molecules, 2024, doi:10.3390/molecules29112497_

Round 1

Reviewer 1 Report

Comments and Suggestions for Authors

In the present article titled “Design, synthesis and photophysical properties of 5-aminobiphenyl substituted [1,2,4]triazolo[4,3-c]- and [1,2,4]triazolo[1,5-c]quinazolines”, Charushin and co-workers explored the preparation triazole-bearing quinazolines from 2-(4-bromophenyl)-4-hydrazino-quinazoline and evaluated their photophysical properties in acetonitrile and toluene. Additionally, the authors investigated the effect of solvent polarity on absorption/emission and performed electrochemical studies and DFT calculations. Although the employed synthetic strategy lacks novelty considering the previous reports of such a strategy (10.1134/S0012500822600298 and 10.3390/molecules28041937), the new obtained quinazolines seem to complement the ones studied before by the same group. Overall, the manuscript is well written. Therefore, I would like to recommend the submitted article for publication.

Author Response

Thank you so much for your comments. Indeed, the current work represents the continuation of research conducted by our scientific group in 2022 an 2023.

Reviewer 2 Report

Comments and Suggestions for Authors

The manuscript submitted for evaluation is entitled: ‘Design, synthesis and photophysical properties of 5-aminobiphenyl substituted [1,2,4]triazolo[4,3-c]-and 3 [1,2,4]triazolo[1,5-c]quinazolines’, is of scientific value, the introduction is adequate to the title and area of research, the experiments are correctly described and carried out. The manuscript can be published on Molecules after several errors are clarified:

The authors should correct the description of the MS spectra, the masses of the M+2 peaks are 1 higher than the given molecular masses of the substance - the authors probably gave Mol. Wt. not Exact Mass. In this case, the latter value should be provided. The authors should also correct the shift range on line 536 (it is 7.75-7.75).

The authors should also improve the recording of the yield of compounds 3a and 3b - it should be in the same format as for compounds 2a and 2b and not above the reaction arrow.

Author Response

- Indeed, Mol. Wt. was given in initial version, we provided Exact Mass in the corrected version. Also we corrected the range in the line 536 as 7.75-7.85.

- We modified the scheme 1 and added the yield of compounds 3a and 3b below the structure, specifying yields for route 1 or 2.

Reviewer 3 Report

Comments and Suggestions for Authors

The authors reported design, synthesis and photophysical properties of [1,2,4]triazolo[4,3-c]- and 3 [1,2,4]triazolo[1,5-c]quinazolines. These results are of help in molecular design for fluorescent molecules. I recommend this manuscript to be published after considering the following points.

 In Table 1, there are no results about solid-state fluorescence properties for 4c, so the authors should add the results.

 Triazolo[4,3-c]quinazolines 2a,b were isomerized to triazolo[1,5-c]quinazolines 3a,b by heating in acetic acid.

The authors should add this reaction mechanism to the supplementary information.

Author Response

  • The solid-state fluorescence of compound 4c was very poor, we added in the table the corresponding information.
  • The proposed reaction mechanism was added to the Supplementary information.